Journal of
**open** psychology data

# Preliminary Data from the Small World of Singlish Words Project: Examining Responses to Common Singlish Words

**DATA PAPER**

**JIN JYE WONG** 🆔

**CYNTHIA S. Q. SIEW** 🆔

*Author affiliations can be found in the back matter of this article

## ABSTRACT

We report data from the Small World of Singlish Words (SWOSW) Project, where word associations are obtained for a list of common Singlish words. Singapore English, colloquially known as Singlish, is a dialect of English spoken in Singapore. Free association data on these words were collected from a young adult population of university undergraduates using an online survey platform. This data enables the construction of semantic networks, allowing one to examine the semantic structure of individual words in the Singlish lexicon, as well as to compare differences in semantic structure across groups of participants.

**CORRESPONDING AUTHOR:**
**Jin Jye Wong**
National University of Singapore, Singapore
jin_jye@nus.edu.sg

**KEYWORDS:**
Singlish; Singapore English; free associations; semantic networks

**TO CITE THIS ARTICLE:**
Wong, J. J., & Siew, C. S. Q. (2024). Preliminary Data from the Small World of Singlish Words Project: Examining Responses to Common Singlish Words. *Journal of Open Psychology Data*, 12: 3, pp. 1–5. DOI: https://doi.org/10.5334/jopd.108

# (1) BACKGROUND

As an extension to ongoing efforts to obtain word association norms for several languages in a large online citizen-science projects such as the Small World of Words project (SWOW; De Deyne et al., 2019), the Small World of Singlish Words aims to collect word associations from native speakers of Singapore English (colloquially known as Singlish) in order to elicit a large-scale free association network of Singapore English. Singlish is an English-based creole language that differs from standard English at both the lexical and grammatical levels (Leimgruber, 2011) and incorporates elements of other languages commonly spoken in Singapore such as Hokkien, Cantonese, Malay, and Tamil, among others (Platt, 1975).

Such word associations provide us with important information about mental representations. It has been shown that the structure of the mental lexicon affects cognitive processing at both the macro- and micro-level (Kenett et al., 2017; Vitevitch, Chan, & Roodenrys, 2012). The construction of a free association network of Singlish words will thus let us further examine the properties of said words at both global and word levels. This network of an English creole can also be compared with a standard English network.

# (2) METHODS

## 2.1 STUDY DESIGN
### Procedure
Data collection was conducted on Qualtrics, an online survey platform. Participants first filled in their demographic information, which includes gender, age, ethnicity, birth country, and spoken languages. Next, participants were presented with a number of Singlish cue words (or short phrases) one at a time, randomly selected from a master cue sheet known as *wiki-300* (see section 2.5, Materials, for more details), and were tasked to provide up to three responses to each cue via input using a keyboard. Participants were initially shown a total of 40 cues, and the number of cue words shown to each participant was gradually increased to 70 until the total task completion time was approximately half an hour, with the final mean task completion time being 28.2 minutes (*SD* = 73.9).

Participants were also able to indicate, using checkboxes, that they (i) did not recognize the word (and were thus unable to provide any responses), or (ii) recognized the word but were unable to provide any responses. Participants were not allowed to proceed to the next word until either a minimum of one response was given or one checkbox was marked.

The specific instructions provided to the participants before the presentation of cues are as follows: "On the top of the screen, a Singapore English word will appear. Enter the *first word that comes to mind* when reading this word. Click on the respective blanks to add a second and third word or proceed to the next word if you can't think of any. If you don't know the word at all, you can just proceed to the next word." On the same page, participants were also provided with two guidelines to "Only give associations to the word on top of the screen (not to your previous responses!)" and "Try to avoid full sentences or long phrases as responses".

## 2.2 TIME OF DATA COLLECTION
Data collection was conducted from March 2022 to May 2022.

## 2.3 LOCATION OF DATA COLLECTION
Data collection was conducted entirely online, using the online survey platform Qualtrics.

## 2.4 SAMPLING, SAMPLE AND DATA COLLECTION
### Participants
Participants were sourced from the National University of Singapore's psychology research pool, which consisted of undergraduates currently taking introductory psychology courses. A single person could participate multiple times as they would receive a different randomized sample of the cues each time, and participants received course credit for participation.

The final dataset consists of 610 participants, of whom 388 (63.6%) identified as female. The average age was 21.8 years (*SD* = 1.95). Besides gender and age, information about race, birth country, and spoken languages were also collected. Regarding ethinicity, 569 (93.2%) identified as Chinese, 21 (3.4%) identified as Malay, and 11 (1.8%) identified as Indian, while the remainder consisted of various other races. Most participants indicated their birth country as either Singapore (*n* = 537, 88.0%) or Malaysia (*n* = 40, 6.6%); other birth countries include China, India, Brunei Darussalam, Indonesia, Philippines, Tajikistan, Myanmar, Australia, and the United States of America.

Participants were also asked if they were native (L1) English speakers, and 588 (96.4%) indicated that they were, while the remainder were not. Other languages spoken by participants include Chinese (*n* = 570, 93.4%), Malay (*n* = 58, 9.5%), and Tamil (*n* = 16, 2.6%), where the distribution generally followed that of ethnicity. Participants could indicate more than one non-English language spoken. Apart from the above three languages, other non-English languages and dialects spoken by participants include Bahasa Indonesia, Baba Malay, Cantonese, Teochew, Hokkien, Hindi, Gujarati, Bengali, Saurashtra, Burmese, Japanese, Korean, Thai, French, German, Spanish, Italian, and Tagalog.

Regarding participant responses, we aimed for each cue to be responded to by at least 100 participants,

following De Deyne et al. (2015), to ensure that a network of sufficient quality could be constructed. In the final dataset, each cue was responded to by an average of 101.8 participants (*SD* = 7.92), and each participant provided at least one response to 49.8 cue words (*SD* = 10.2).

## 2.5 MATERIALS

The set of Singlish cues used in the study comprised of the words on the Wikipedia page for Singlish Vocabulary (*Singlish vocabulary*, 2022), under the section "List of Singlish Words". This set of words were chosen as Wikipedia is a crowd-sourced reference site; hence, we reasoned that these were lexical items that were likely to be well-known by, and generally familiar to, Singaporeans. Limiting the number of cues to the most prominent Singlish words also ensures that enough association data could be collected for each individual word, the aim being to obtain responses from approximately a hundred individuals per word in order to later construct an association network of sufficient quality. The words under "Food and beverages" and "English words with different meanings in Singlish", as well as the phrases in "Expressions", were omitted. After curation to remove offensive terms, the final list, which we call *wiki-300*, contained 299 Singlish words.

## 2.6 QUALITY CONTROL
### Data Cleaning

The data was cleaned following a procedure highly similar to the one used by the original Small World of Words project (De Deyne et al., 2019); changes to the procedure were made in order to handle the unique data characteristics of Singlish word associations.

Primarily, a significant proportion of responses were verbose (i.e. consisted of more than one word). In such cases, a *splitting* approach was used, where multi-word responses were split into their constituent words. For instance, given the cue phrase *Ah Beng*, if a participant were to respond with the phrase *young hooligan*, they would be recorded as providing two separate responses, *young* and *hooligan*. The main drawback to this approach was that each participant now contributes unevenly to the data, since providing a multi-word response would lead to each individual word in that response becoming individual responses. However, this was deemed acceptable as participants did not already contribute uniformly to the network at the outset of data collection (e.g. a single person could participate multiple times).

For spelling correction, the *hunspell* R package (Ooms, 2020) was used. A custom Singlish dictionary was created by adding *wiki-300* to the default British English dictionary. Since many cues consisted of more than one word, the individual component words of each phrase were added manually as separate entries. For example, the cues *buay pai* ("not bad") and *buay tahan* ("can't endure") would result in the monograms *buay, pai,* and

*tahan* being added. In the automated pipeline, words deemed to be misspelled were automatically replaced with the first word suggested by *hunspell*, with the function *hunspell_suggest*. This also entails that words spelt differently in American English would be corrected to their British English equivalents; consequently, different spellings of a word (e.g. *color* and *colour*) are treated as the same word (*colour*). A total of 3266 (3.6%) responses underwent spelling correction.

To convert words to their standard forms, lemmatization was performed using the *lemmatize_words* function from the *textstem* R package (Rinker, 2018). This maps inflections of any word back to their root word (e.g. *incurred* and *incurs* would be corrected to *incur*). Finally, any remaining word-final punctuation, such as full stops, tags, and quotes, was removed. 41 instances of remaining non-English characters, all determined to be Mandarin characters, were removed manually.

# (3) DATASET DESCRIPTION AND ACCESS

The data was published on the Open Science Framework on 15th May 2023 and follow the FAIR guidelines (Wilkinson et al., 2016). The data are licensed under Creative Commons Attribution 4.0 International (CC BY 4.0). The table below provides an overview of the different files containing the data. All files are in English. Data files are comma-separated values (.csv) files.

| FILE | DESCRIPTION |
| --- | --- |
| wiki_300_cue_list.csv | Contains the Singlish cues used in the study, as well as short definitions of each term. |
| participant_information.csv | Contains the details of each participant. |
| cleaned_responses_long.csv | Contains responses of participants in a long format, where each row contains one participant's responses to a single cue. |
| codebook.txt | Provides descriptions of column titles in data files. |
| full_data_raw.csv | Contains uncleaned responses of each participant in a wide format. |
| Dictionary files | A folder containing the custom Singlish dictionary mentioned in Section 2.6 (Quality Control), meant to be used with *hunspell*. Both files can be opened with a regular text editor. |

## ETHICAL CONSIDERATIONS

All data were recorded in reference to a randomly generated identifier comprising of four digits, assigned to participants at the beginning of the study. Identifying information such as names, addresses, date of birth, or occupation were not recorded. Participants provided

informed consent that their anonymised data could be shared with other parties for research purposes. This study was approved by the Department Ethics Review Board of the Department of Psychology at the National University.

## REUSE POTENTIAL

The reported data present the first ever large-scale collection of free association responses to Singlish lexical items, a unique dataset with no predecessor. The data can be reused in multiple ways.

As an English-based creole, language examination of the structure of the Singlish lexicon can lead to further insights about other English-based creoles. For a start, the structural properties of a Singlish network derived from this dataset can be compared to the larger association network produced by the Small World of Words project (De Deyne et al., 2019) that contains purely English words, in order to examine if there are any preliminary differences in structure. Following this, this dataset also serves as a springboard to examine other English creoles such as Hawaiian creole (Odo, 1970), or even other Asian languages such as Mandarin, given that the composition of Singlish includes elements from Chinese, Malay, Tamil, and other Asian languages.

The collected data can also be used to augment language models for use in natural language processing (NLP). Prior work by De Deyne, Perfors and Navarro (2016) has showed that a model trained on association data alone can outperform traditional NLP models trained on text corpora in the task of word similarity judgements. Models augmented with knowledge graphs, which are effectively word association networks with labelled edges for relation types, also outperform baseline language models on commonsense reasoning tasks (Liu, Cohn, and Frermann, 2021). Word association data can thus supplement existing language models, especially in cases where language resources are scarce. This is all the more crucial for Singlish, given the lack of NLP tools available for analysing it, as noted by Hsieh et al. (2022).

Finally, the portable and well-bounded nature of this dataset also makes it suitable for educational purposes in teaching students various aspects of handling word association data, such as performing basic statistical, network, or natural language analyses on programming platforms such as R (Ihaka & Gentleman, 1996) or Python.

Given that this is the first collection of Singlish free association data, there are several avenues for improvement, two of which are the most significant. First, the data was collected from a sample of university students, resulting in a limited age range and a particular demographic (i.e., predominantly Chinese participants who are over-represented in our dataset at 93.4%, relative to our national distribution of 74.1%; Singapore Department of Statistics, 2022). Second, responses were only collected for a limited set of Singlish words which represent only a small portion of the entire Singlish lexicon.

Further studies are ongoing which aim to collect responses from the wider Singaporean population, particularly involving Singaporeans from non-majority ethnic groups with differing levels of educational attainment and of different ages. A more extensive list of cues that include Singlish terms from more sources, such as Singlish dictionaries like the Coxford Singlish Dictionary (Goh & Woo, 2009), has also been prepared. The qualities of an association network constructed from responses involving both the wider Singaporean population and a more extensive cue list was presented in Wong and Siew (2023), though a greater number of participants is required per cue to ensure better data quality.

## FUNDING INFORMATION

This project was funded by a grant from the DSO National Laboratories ("Network model of Singlish Words", DSOCL20125, Years active: 2021–2023).

## COMPETING INTERESTS

The authors have no competing interests to declare.

## AUTHOR CONTRIBUTIONS

Anutra Guru (now at King's College London) and Chester Tan (now at Julius-Maximilians-Universität Würzburg) helped to curate the cue list. Jazton Chern assisted in the literature review. Wong Jin Jye: Data curation, Methodology, Project administration, Writing – Original draft. Cynthia S. Q. Siew: Conceptualization, Methodology, Supervision, Funding Acquisition, Writing – review & editing.

## AUTHOR AFFILIATIONS

**Jin Jye Wong** (iD) orcid.org/0009-0007-3248-4942
National University of Singapore, Singapore
**Cynthia S. Q. Siew** (iD) orcid.org/0000-0003-3384-7374
National University of Singapore, Singapore

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

## PEER REVIEW COMMENTS

*Journal of Open Psychology Data* has blind peer review, which is unblinded upon article acceptance. The editorial history of this article can be downloaded here:

- **PR File 1.** Peer Review History. DOI: https://doi.org/10.5334/jopd.108.pr1

**TO CITE THIS ARTICLE:**
Wong, J. J., & Siew, C. S. Q. (2024). Preliminary Data from the Small World of Singlish Words Project: Examining Responses to Common Singlish Words. *Journal of Open Psychology Data,* 12: 3, pp. 1–5. DOI: https://doi.org/10.5334/jopd.108

**Submitted:** 30 January 2024     **Accepted:** 03 April 2024     **Published:** 18 April 2024

