## [Peer Review History. · Journal of Open Psychology Data]

Reviewer A:

Recommendation: Revisions Required

Comments to the author(s)

The contents of the paper and the deposited data fulfilled most of the requirements, except for the following two:

1. In "Dataset description and access" the author mention that "The table below provides an overview of the different files and folders containing raw data, cleaned data, and the scripts and resources used in cleaning of the data.", but the OSF repo doesn't have any scripts. Also, it's not clear what exactly do they mean by 'resources' here? There are only .csv file, a PDF and a folder with two other files.

The paragraph has been reworded to avoid ambiguity.

2. The folder 'Dictionary files' contain two files ('en_singlish_custom.aff' and 'en_singlish_custom.dic'), but these files don't fulfil the requirement that "The deposited data must be actionable – i.e. if a specific script or software is needed to interpret it, this should also be archived and accessible.". I couldn't open these files.

The dictionary files can be opened using a regular text editor (e.g. Notepad, if using Windows).

There is on type (extra 'both'): cognitive processing, both at *both* the macro- and micro-level

After above small changes the submission can be accepted.

Reviewer H:

Recommendation: Revisions Required

Comments to the author(s)

This paper accompanies a word association data set for 299 words of Singapore English (Singlish). The data are openly available in an accessible manner and the procedures that were used are well documented. I appreciate the fact that the authors included the verbatim instructions.

I have a few suggestions for improving the accompanying text so that end-users may make the most of these data:

1. The title indicates that the data are preliminary. In what way do the authors expect to expand upon the existing data (e.g., add cue words, include participants with different background characteristics, ...)?

Greater detail as to how the existing data is intended to be expanded has been added to the end of the section on Reuse Potential.

2. When the authors suggest that the properties of novel Singlish words can be examined (background), what do they mean exactly by 'novel'? Do they mean less common Singlish words?

The original intent was to convey that these Singlish words have not been studied in detail previously. The word 'novel' has been removed to avoid ambiguity.

3. I appreciate the fact that the authors ensured that at least 100 participants provided a response to each cue so that a network of sufficient quality can be constructed. This is often overlooked. I am not in the habit of suggesting that authors include a reference to my own work, but in case the authors would find it worthwhile, they might want to mention De Deyne et al. (2015), which provides the rationale for this sample size.

The suggested reference has been included.

4. Participants were presented with 40-70 cue words (procedure). How was the number of cue words per participant determined?

The number of cue words per participant were adjusted based on task completion times. Additional detail has been added to the Procedure section.

5. The participants section mentions 608 participants, while the participants data file has 610 entries. Where does this discrepancy come from?

The smaller number was a holdover from an earlier draft that used an older dataset. This discrepancy and other consequent changes to participant information have been corrected.

6. If they exist, I would recommend providing (approximate) English translations along the Singlish cue words in the wiki 300 cue list to facilitate the data's uptake by international scholars. I would also be interested in learning if there are any domains which are predominantly covered by these words to better understand what the resulting network would be a representation of.

The wiki-300 cue list has been updated to include a column where approximate definitions have been provided. I am unsure what is referred to by "domains", but Singlish is predominantly used in an informal context, and the terms themselves refer to a wide variety of concepts including, but not limited to, foods, emotions, dispositions, and particular archetypes/personalities.

7. The data cleaning is described in detail but as a potential end-user I would be interested in learning what percentage of the data was affected by the various procedures. For instance, what percentage of associations underwent automatic spelling correction, how many non-English entries were removed, ...?

The section on Data Cleaning has been updated to add greater detail.

8. The dataset description mentions raw data and scripts used for data cleaning, but I did not find these on the osf. I could only retrieve the cleaned data and the custom dictionaries that were used.

The paragraph has been reworded to avoid ambiguity.

9. Not all in-text references are included in the reference list (e.g., Kenett et al., 2017; Vitevitch et al., 2012, ...)

The list of references has been updated.

References:

De Deyne, S., Verheyen, S., & Storms, G. (2015). The role of corpus size and syntax in deriving lexico-semantic representations for a wide range of concepts. *The Quarterly Journal of Experimental Psychology*, 68, 1643-1664.

I always sign my reviews,
Steven Verheyen
Erasmus University Rotterdam